

# Co-differential genes between DKD and aging: implications for a diagnostic model of DKD

Hongxuan Du[1,2,*], Kaiying He[1,2,*], Jing Zhao[3], Qicai You[1,2], Xiaochun Zhou[2] and Jianqin Wang[1,2]

[1] Lanzhou University, Lanzhou, Gansu, China
[2] Department of Nephrology, The Second Hospital of Lanzhou University, Lanzhou, Gansu, China
[3] Department of Pediatric Cardiology, nephrology, rheumatism and Immunology, Gansu Provincial Central Hospital, Lanzhou, Gansu, China
[*] These authors contributed equally to this work.

Corresponding authors
Xiaochun Zhou,
ery_zhouxc@lzu.edu.cn
Jianqin Wang,
ery_wangjqery@lzu.edu.cn

## ABSTRACT

**Objective**. Diabetic kidney disease (DKD) is a serious complication of diabetes mellitus (DM) that is closely related to aging. In this study, we found co-differential genes between DKD and aging and established a diagnostic model of DKD based on these genes.

**Methods**. Differentially expressed genes (DEGs) in DKD were screened using GEO datasets. The intersection of the DEGs of DKD and aging-related genes revealed DKD and aging co-differential genes. Based on this, a genetic diagnostic model for DKD was constructed using LASSO regression. The characteristics of these genes were investigated using consensus clustering, WGCNA, functional enrichment, and immune cell infiltration. Finally, the expression of diagnostic model genes was analyzed using single-cell RNA sequencing (scRNA-seq) in DKD mice (model constructed by streptozotocin (STZ) injection and confirmed by tissue section staining).

**Results**. First, there were 159 common differential genes between DKD and aging, 15 of which were significant. These co-differential genes were involved in stress, glucolipid metabolism, and immunological functions. Second, a genetic diagnostic model (including IGF1, CETP, PCK1, FOS, and HSPA1A) was developed based on these genes. Validation of these model genes in scRNA-seq data revealed statistically significant variations in FOS, HSPA1A, and PCK1 gene expression between the early DKD and control groups. Validation of these model genes in the kidneys of DKD mice revealed that Igf1, Fos, Pck1, and Hspa1a had lower expression in DKD mice, with Igf1 expression being statistically significant.

**Conclusion**. Our findings suggest that DKD and aging co-differential genes are significant in DKD diagnosis, providing a theoretical basis for novel research directions on DKD.

## INTRODUCTION

Diabetic kidney disease (DKD) is a severe complication of diabetes mellitus (DM). Over the past two decades, the incidence and mortality rates of this disease have increased rapidly (*Xiong & Zhou, 2019*). In addition, the prognosis is poor. DKD is a principal factor in the development of end-stage renal disease (*Atkins & Zimmet, 2010*). DKD is responsible for 34%–36% of fatalities among various types of chronic kidney diseases (*Thomas, 2019*). However, the understanding of the pathogenesis of DKD remains incomplete, and a definitive cure is yet to be discovered. Therefore, a deeper understanding of its pathogenesis and treatment methods is necessary to enhance its diagnosis and therapy.

Studies have indicated that aging is a significant contributor to the development of DKD, and can increase the prevalence of nephrosclerosis (*Alicic, Rooney & Tuttle, 2017*). Cellular senescence (cellular aging) is a critical factor in aging (*Xiong & Zhou, 2019*). Hyperglycemia stimulation in both *in vitro* or *in vivo* (DKD rats and mice) experiments has been found to accelerate cellular senescence in various types of renal cells, including proximal tubular epithelial cells (*Kitada et al., 2014*; *Tsai et al., 2018*; *Verzola et al., 2008*), endothelial cells (*D'Onofrio et al., 2016*), mesangial cells (*Cao et al., 2019*), and podocytes (*Verzola et al., 2008*), as well as shorten telomere length (*Verzola et al., 2008*). These findings suggest a strong correlation between aging and DKD.

In addition, hyperglycemia can promote the formation of the senescence-associated secretion phenotype (SASP) (*Prattichizzo et al., 2018*). This refers to the secretome produced by senescent cells. The SASP encompasses interleukins, chemokines, inflammatory factors, and growth factors (*Vicente et al., 2016*). SASP not only promotes cell senescence but also damages the immune system function, resulting in persistent inflammation and fibrosis (*Xiong & Zhou, 2019*). In addition, cellular senescence affects somatic cells, such as kidney cells, and causes cumulative alterations in the immune system. These changes result in immunosenescence (immune aging), which is characterized by a decline in adaptive immunity and an increase in low-level chronic inflammation (*Barbé-Tuana et al., 2020*). Immunosenescence also alters the quantity and functionality of immune cells, such as T and B lymphocytes, macrophages, and mast cells (*Yang & Mou, 2017*).

There is accumulating evidence suggests that the immune system plays a crucial role in DKD, challenging the traditional view that DKD is solely caused by metabolic and hemodynamic changes. The infiltration of immune cells is usually detected in renal samples at various phases of DKD (*Klessens et al., 2017*). Immune cells establish a pro-fibrotic microenvironment for renal cells by releasing cytokines and pro-fibrotic factors (*Pérez-Morales et al., 2019*). Owing to the activation of numerous signaling pathways, such as nuclear factor-kappa B (NF-κB), tumor necrosis factor-β (TGF-β), and mitogen-activated protein kinase (MAPK), chronic and continuing inflammation eventually leads to progressive renal fibrosis (*Tang & Yiu, 2020*). Moreover, some studies have explored the potential protective effects of anti-inflammatory therapy in DKD (*Pérez-Morales et al., 2019*). These findings provide evidence that inflammation promotes DKD.

Therefore, cellular senescence may contribute to DKD by exacerbating immune dysregulation. Immune cell alterations during immunosenescence play a significant role in the pathological progression of DKD (*Barbé-Tuana et al., 2020*).

Therefore, aging and DKD are inextricably linked and both present a similar low-level chronic inflammation state. This suggests a common genetic background between the two. However, few studies have integrated both the genetic background factors. This study aimed to conduct integrative research on type 2 DKD and aging by analyzing their co-differential genes. A genetic diagnostic model of DKD was developed by differentially expressed gene (DEG) analysis (*Ritchie et al., 2015*) and LASSO regression analysis (*Tibshirani, 1997*). Additionally, this study used consensus clustering analysis (*Wilkerson & Hayes, 2010*), WGCNA (*Langfelder & Horvath, 2008*), functional enrichment analysis (*Consortium, 2019*; *Kanehisa & Goto, 2000*), PPI network analysis (*Szklarczyk et al., 2019*), and immune cell infiltration analysis (*Rusk, 2019*) to investigate the biological mechanisms of DKD and aging co-differential genes, with a particular focus on understanding the mechanism of immune cells. Finally, the expression of these co-differential genes was verified using single-cell RNA sequencing (scRNA-seq) data from patients with early DKD and qRT-PCR tests on DKD mouse kidneys.

## MATERIALS & METHODS

### Data mining

The datasets containing gene expression profiles (GSE96804, GSE30528, GSE30529, and GSE131882) were downloaded from the GEO Database (https://www.ncbi.nlm. nih. gov/gds/). GSE96804 was used as an experimental dataset, containing human exon level expression profiles of glomeruli from the kidneys of 41 patients with type 2 DM complicated with DKD and 20 normal controls (*Pan et al., 2018*). The GSE30528 and GSE30529 datasets from the GSE30122 dataset were used for validation. These datasets consist of transcriptome analysis data for human glomeruli and renal tubules (*Woroniecka et al., 2011*). In addition, the GSE30528 and GSE30529 datasets consisted of nine and ten patients with DKD, respectively, and 13 and 12 controls, respectively. The GSE131882 dataset contained scRNA-seq data from three kidney samples of patients with type 2 DM complicated (aged 52, 57, and 74 years) with early DKD, as well as three control samples (aged 54, 61, and 62 years) (*Wilson et al., 2019*). Next, the probes were annotated using gene symbols. The average values of probes were computed when multiple probes matched the same gene symbol. In addition, the Human Genomic Resources website (https://genomics.senescence.info/) provided information on 307 genes that are considered aging-related genes (*Tacutu et al., 2018*).

### DEG analysis

DEGs between patients with DKD and normal controls were assessed on the retrieved gene expression matrix of GSE96804 using the "limma" (version 3.50.1) package in R software (version 4.1.1) (*Ritchie et al., 2015*; *R Core Team, 2021*). Genes with FDR (adjusted *p*-value) < 0.05 were considered as DKD-DEGs. Genes with FDR <0.05 and |LogFC| > 1 were considered as significant DKD-DEGs. The volcano plot of GS96804 was created

and visualized using "ggplot2" R package (version 3.4.4). Next, the co-differential genes between DKD and aging were identified by intersecting DKD-DEGs and aging-related genes. The intersection of significant DKD-DEGs and aging-related genes was used to identify significant co-differential genes in both DKD and aging.

## Gene functional enrichment analysis

We conducted gene set enrichment analysis (GSEA) (*Subramanian et al., 2005*) using the "clusterProfiler" package (version 4.2.2) of the R software to determine whether the aging biological process was enriched between the DKD and control samples (*Yu et al., 2012*). This analysis was conducted using the gene expression data from GSE96804. GO (*Consortium, 2019*) and KEGG (*Kanehisa & Goto, 2000*) enrichment analysis of genes were also performed using "clusterProfiler". After filtration (FDR < 0.05), the relevant elements were visualized using R. Visualization was based on the enrichment score, which was calculated as follows: Enrichment score = GeneRatio/BgRatio. Some KEGG figures were plotted using https://www.bioinformatics.com.cn (*Tang et al., 2023*).

## Consensus clustering analysis

Consensus clustering of DKD samples was conducted using the expression matrix of aging-related genes, which were differentially expressed in DKD patients compared to normal controls. The cluster was operated using the "ConsensusClusterPlus" (version 1.58.0) in R (*Wilkerson & Hayes, 2010*). The process began with the minimum category number ($k = 2$) until the maximum value ($k = 9$) was obtained iteratively and incrementally. After considering various factors, such as the consensus cumulative distribution function (CDF), changes in the area under the CDF curve, tracking plots, and heatmaps of consensus clustering, the optimal parameter can be chosen. DKD samples were categorized into several groups based on their characteristics (aging and DKD co-differential DKD subtypes). Moreover, the MSigDB hallmark gene sets, loaded *via* the "msigdbrs" package (version 7.5.1) in R (*Liberzon et al., 2015*), were used to perform ssGSEA to reveal variations in hallmark pathways in different aging and DKD co-differential DKD subtypes (adjusted *p*-value < 0.05). Subsequently, a heatmap was used to represent the data visually.

## WGCNA co-expression analysis

The DKD gene expression data of samples in GSE96804 were evaluated, and samples were categorized based on consensus clustering of aging and DKD co-differential DKD subtypes. Next, co-expression analysis was conducted on the top 10,000 genes with the highest median absolute deviations to exclude genes with minimal alterations. A co-expression network was constructed using the "WGCNA" package (version 1.70-3) in R. This allowed the identification of co-expression clusters (modules) consisting of closely related genes. The eigengene of the module with phenotypes (aging and DKD co-differential DKD subtypes) was analyzed using eigengene network methodology (*Langfelder & Horvath, 2008*). Subsequently, the key genes were summarized (screened by GS > 0.7 and MM > 0.8).

## Immune cell infiltration analysis

The "CIBERSORTx" (https://cibersortx.stanford.edu/) is an online analytical tool for estimating the relative fraction of different immunocytes in blended cell colonies (*Rusk, 2019*). This tool was used to analyze the correlation between immune cell infiltration and aging-related DEGs in DKD. This analysis was conducted using the LM22 dataset, which contains data on 22 human hematopoietic cell phenotypes.

## Protein-protein interaction networks (PPI)

PPI (*Szklarczyk et al., 2019*) network analysis was conducted on the genes using the "STRING" platform (https://cn.string-db.org/). The analysis was performed with medium confidence (Interaction Score > 0.4).

## Pearson's correlation test

The Pearson's correlation test was performed on GSE96804 gene expression data using "hmisc" package in R (version 5.1-1) to identify the co-differential genes associated with DKD and aging.

## LASSO regression analysis to develop diagnosis model

The LASSO regression analysis (*Tibshirani, 1997*) was conducted using the "glmnet" (version 4.1-4) package in R (*Engebretsen & Bohlin, 2019*) to develop aging-related diagnostic gene markers for DKD. The training set consisted of the expression of aging-related DKD differential genes in the GSE96804 database. The minimum partial likelihood deviance was determined as lambda using ten-fold cross-verification. Next, the GSE30528 and GSE30529 datasets were merged as a validation set to verify the diagnosis model and calculate its diagnostic effectiveness in distinguishing DKD from normal patients. The receiver operator characteristic (ROC) curve was plotted using "ROCR" (version 1.0-11) package in R (*Sing et al., 2005*). Afterwards, AUC was calculated.

## Single cell analysis

The "Seurat" package (version 4.3) in R was used for GSE131882 data analysis. Screening and analysis were conducted on cells with ≥ 500 spotted genes and genes with ≥3 covered cells. Data was standardized using the "SCTransform". UMAP was used to reduce dimensionality and visualize the scRNA-seq data. Subsequently, cluster cell subtypes were annotated based on renal cell type-specific differentially expressed genes (*Wilson et al., 2019*). Gene expression was analyzed and visualized using violin plots.

## Animal models

The C57BL/6J mouse strain is well known for its sensitivity to metabolic syndromes. It is suitable for the establishment of a mouse model of DKD *via* intraperitoneal injection of streptozotocin (STZ). To minimize the number of experimental animals, six pathogen-free wild-type male C57BL/6J mice (age: 6–8 weeks, weight: 19 ± 1 g) were purchased from Lanzhou Animal Research. Mice were randomly allocated to two groups using a simple randomization method. The study on mice was approved by The Second Hospital of Lanzhou University Institutional Ethical Committee (D2019-149) and strictly followed the guidelines established by the Care and Use Guide of Laboratory Animals of the
National Research Council (US) Committee and the Regulations of Laboratory Animals of China. The mice were kept in a controlled environment in the laboratory, with a ambient temperature of $(20 \pm 2)$ °C, ambient humidity of 40–70%, besides a 12 h light/dark cycle. Specific pathogen-free food and water were provided every two days. Mice in the DKD group were fed a high-fat-diet. At ten-weeks-old, the right kidney of the DKD mice were removed after administering tribromoethanol anesthesia. At 12 weeks of age, STZ at 50 mg/(Kg bw)/day was administered intraperitoneally for five days. The established criteria for euthanizing DKD mice involved administering an anesthesia overdose before the end of the experiment. This was based on the condition that, once the mice reached 22 weeks of age, the fasting blood glucose of their tail vein consistently exceeded $\geq 11.1$ mmol/L more than three times. Mice in control group were fed a standard-diet and euthanized at 22 weeks of age using an overdose of anesthesia. At 12 weeks of age, control mice received daily intraperitoneal administration of normal saline solution for five days. After confirming that the mice experienced cardiopulmonary arrest, kidney tissues were promptly removed and rapidly frozen in liquid nitrogen for at least 10 min. Subsequently, they were stored in a refrigerator at $-80$ °C.

## Histological and histopathological analyses

Mouse kidneys were fixed in 4% paraformaldehyde, embedded in paraffin, then cut into 4 μm-thick histological sections. These sections were subsequently stained with hematoxylin and eosin (H&E), Masson's Trichrome, and Periodic Acid-Schiff (PAS) using the SolarBio kit.

## Quantitative reverse-transcription PCR (qRT-PCR)

Approximately 50 mg of the kidney tissue was collected from each sample. The tissue was then mixed with TRIzol reagent (Invitrogen) on ice and pulverized using a tissue grinder. RNA extraction was performed using chloroform, isopropanol, and 80% ethanol. Ultraviolet spectrophotometry was used for nucleic acid quantification, and A260/A280 was measured. Reverse transcription was completed by the GoScript Reverse Transcription System (Promega A5000). A 20 ul reaction volume containing 1 ug RNA was used to complete the reverse transcription at 42 °C for 15 min, followed by 70 °C for 15 min. Enzyme-free EP tubes and pipette tips were used throughout the procedure. Then, qRT-PCR was performed on the ABI7500 system using the qPCR Kits GoTaq® qPCR and qRT-PCR Systems of Promega (A6001). β-actin was used as the control for normalization. The annealing temperature was set at 60 °C. Primer sequences are shown in Table 1.

## Statistical analysis

Data were analyzed using R (version 4.1.1; *R Core Team, 2021*) and SPSS software (version 26). The qRT-PCR data were analyzed using the Kolmogorov–Smirnov test to determine if the data followed a normal distribution. Homogeneity test of variance was also performed. Student's $t$-test or Mann–Whitney test was used based on these outcomes. Differences were considered statistically significant at $p < 0.05$.

**Table 1  Primer sequences information of RT-qPCR.**

| Gene | Forward/Reverse | sequences |
| --- | --- | --- |
| Fos | Forward | ATGGTGAAGACCGTGTCAGG |
|  | Reverse | GTTGATCTGTCTCCGCTTGGA |
| Pck1 | Forward | CGCAAGCTGAAGAAATATGACA |
|  | Reverse | GATGACTGTCTTGCTTTCGATC |
| Igf1 | Forward | GAGGGGCTTTTACTTCAACAAG |
|  | Reverse | TACATCTCCAGTCTCCTCAGAT |
| Hspa1a | Forward | GGTGCTGACGAAGATGAAGGAGATC |
|  | Reverse | CTGCCGCTGAGAGTCGTTGAAG |
| β-actin | Forward | CTACCTCATGAAGATCCTGACC |
|  | Reverse | CACAGCTTCTCTTTGATGTCAC |

# RESULTS

**Gene expression analysis and enrichment analysis of DKD *vs* control**

GSEA of all genes in the GSE96804 dataset revealed that aging-related pathways were significantly enriched in DKD samples compared to control samples (Figs. 1A–1B). A total of 12,245 genes with FDR < 0.05 were identified as DKD-DEGs when compared to the control group. DKD-DEGs contained 6,369 upregulated and 5,876 downregulated genes. Genes with FDR < 0.05 and |LogFC| > 1 were identified as significant DKD-DEGs, with 306 significantly upregulated and 338 significantly downregulated DKD-DEGs. Genes were visualized using a volcano plot (Fig. 1C). Each gene was represented by a dot, and significant DKD-DEGs were colored in red or blue. A total of 159 DKD and aging co-differential genes were identified by the intersection of 12245 DKD-DEGs and 307 aging-related genes (Fig. 1D). GO enrichment analysis revealed that biological processes associated with aging (cellular senescence, replicative senescence, regulation of cellular senescence, positive regulation of cellular senescence, and negative regulation of cellular senescence), stress (response to oxidative stress and cellular response to chemical stress), apoptosis (regulation of apoptotic signaling pathway), and nutritional metabolism (regulation of carbohydrate metabolic process and response to nutrient levels) were enriched by the co-differential genes associated with DKD and aging (Fig. 1E). The minimum enrichment score for these biological processes was 7.33. The maximum enrichment score was 56.92 (replicative senescence). KEGG enrichment analysis also revealed pathways related to aging (cellular senescence, longevity regulating pathway-multiple species, longevity regulating pathway), apoptosis, metabolism (insulin signaling pathway, insulin signaling pathway, insulin resistance, endocrine resistance, growth hormone synthesis, lipid and atherosclerosis, fluid shear stress and atherosclerosis, neurotrophin signaling pathway, adipocytokine signaling pathway), and several signaling pathways related to inflammation, apoptosis or metabolism (FoxO, MAPK, PI3K-Akt, AGE-RAGE, HIF-1, Ras, ErbB, and P53) (Fig. 1F).

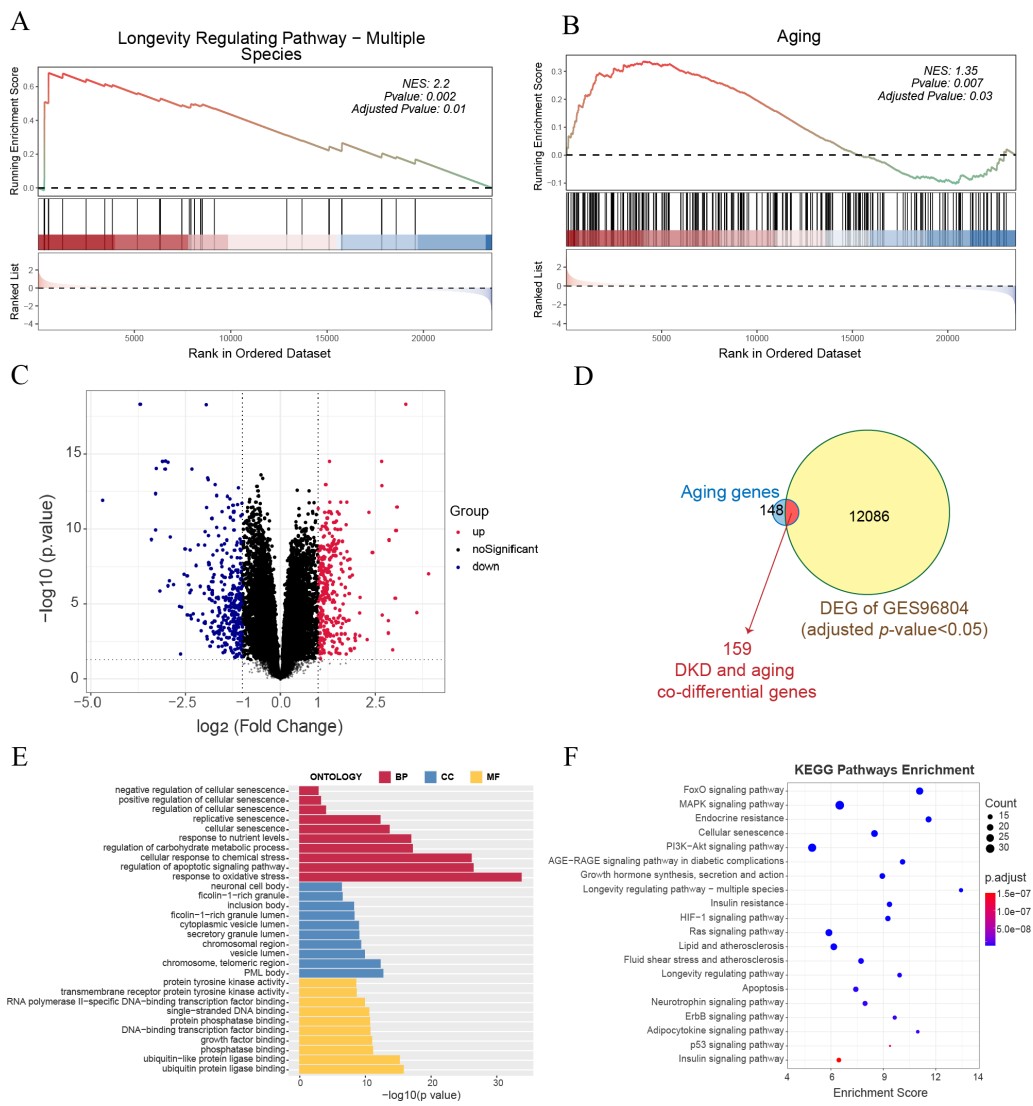

**Figure 1** Gene expression and enrichment analysis for diabetic kidney disease (DKD) *vs* healthy control in GSE96804. (A–B) Gene set enrichment analysis (GSEA) results for all gene expressions. (A) Kyoto Encyclopedia of Genes and Genomes (KEGG) GSEA (longevity regulating pathway—multiple species). (B) ene Ontology (GO) GSEA (Aging). (C) Volcano plot: red represents significantly upregulated differentially expressed genes (DEGs), and blue represents significantly downregulated DEGs. (Above the dotted line, *p*-value < 0.05) (D) Venn diagram shows the 12,245 DEGs intersected with 307 aging-related genes. (E–F) GO analysis of biological process (BP), cellular component (CC), and molecular function (MF), and KEGG pathway results enriched by DKD and aging co-differential genes.

## Characterization of aging and DKD co-differential DKD subtypes

Consensus clustering analysis was conducted to cluster DKD samples into three aging and DKD co-differential DKD subtypes (groups 1–3), based on the expression of 159 co-differential genes that were associated with DKD and aging in 41 DKD samples (Figs. 2A–2D). The three DKD subtypes were labeled as group 1 ($n = 20$), group 2 ($n = 19$), and group 3 ($n = 2$), which had distinct gene expression patterns of co-differential genes. Analysis

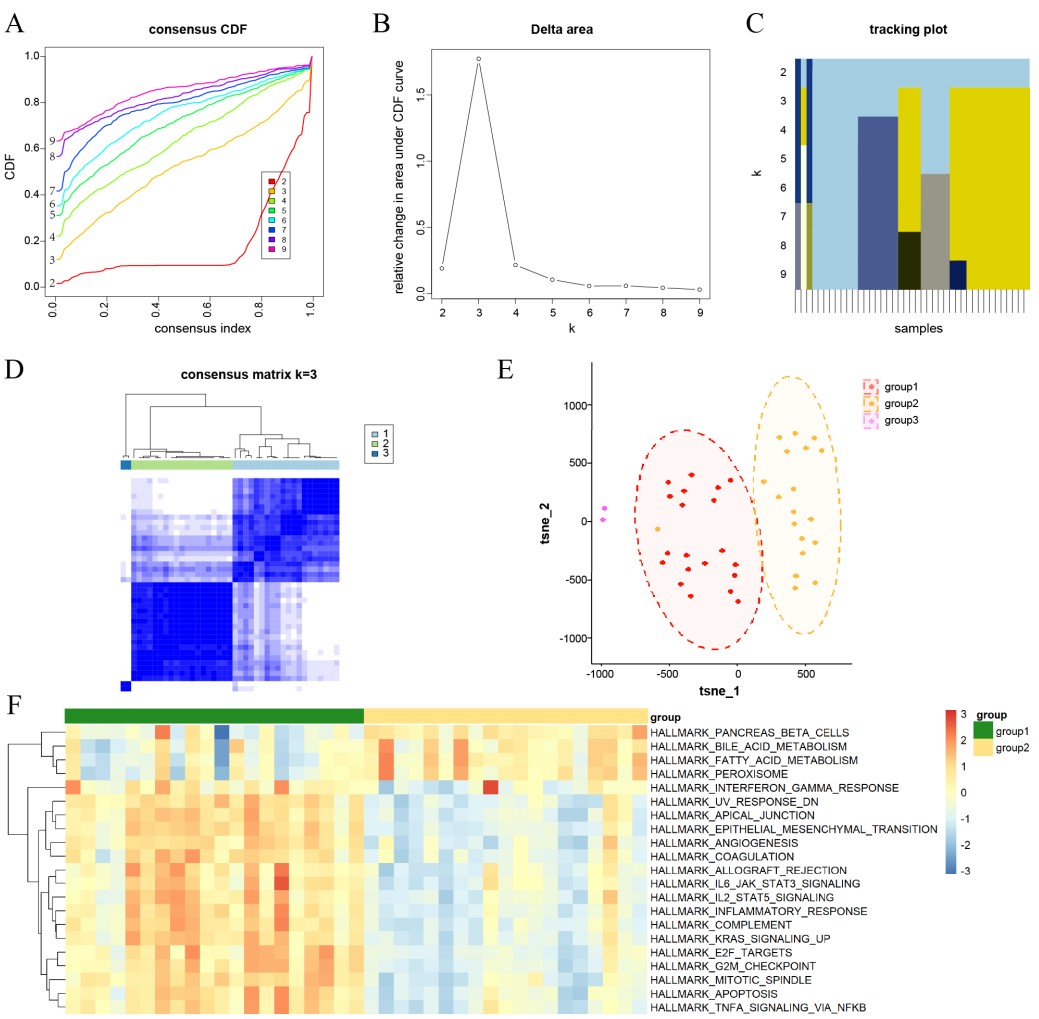

**Figure 2 Consensus clustering analysis results for DKD samples and their characterization.** (A) Consensus cumulative distribution function (CDF) plots for every k. (B) Line chart of the relative alterations in the areas under the CDF curves. (C) Tracking plots displaying consensus clusters at every k. (D) Consensus matrix plots representing the consensus values when $k = 3$. (E) t-SNE plots confirm the classification accuracy of the three aging and DKD co-differential DKD subtypes. (F) Heatmap of the differences in the enrichment score of hallmark pathways between groups 1 and 2 by ssGSEA (red represents upregulation and blue represents downregulation).

of the t-SNE plots revealed significant variances in whole-genome expression among the samples from the three groups (Fig. 2E). The sample size of group 3 was insufficient; hence, it was not included in the subsequent study. In groups 1 and 2, the expression levels of significant co-differential genes associated with DKD and aging showed marked heterogeneity (Fig. 2F). Furthermore, different subtypes exhibited distinct differences in several biological processes associated with DKD development. Group 1 exhibited lower metabolism of bile acids and fatty acids and downregulation of genes encoding pancreatic beta cells and peroxisomes than group 2. In contrast, the genes encoding for inflammatory

response, apoptosis, angiogenesis, cell cycle, and epithelial-mesenchymal transition were upregulated (Fig. 2F).

## Identification of co-expressed gene modules

A WGCNA co-expression network was established by analyzing all 10,000 gene expressions from 41 DKD samples in GSE96804. The dataset was verified and two missing values were identified and removed (cut height = 95) (Fig. 3A). All missing values belonged to group 3 of the aging and DKD co-differential DKD subtypes. After removing them, no members of group 3 remained. The sample dendrogram was clustered using 10,000 gene expression profiles. However, as shown in Fig. 3B, the dendrogram result was identical to the consensus clustering result, which was based only on 159 DKD and aging co-differential genes. As shown in Fig. 3C, we set 3 as the power of soft-thresholding. Co-expression modules were created using dynamic tree cut. Subsequently, at a height cut of 0.25, 11 modules were merged into 10 modules and unique colors were assigned (Fig. 3D). As shown in Fig. 3E, the turquoise module was most associated with aging and DKD co-differential DKD subtypes, implying that genes in the turquoise module were highly involved in the grouping of co-differential DKD subtypes. The features of the turquoise module genes are shown in Fig. S1. A total of 49.58% of the turquoise module genes were DKD-DEGs. 73 turquoise module genes were aging-related, accounting for 23.78% of all 307 aging-related genes. Five genes (FOS, EGR1, PTGS2, IGF1, and PLCG2) were identified from the intersection of the turquoise module genes and significant DKD-DEGs.

## Study of key genes in co-expression gene module

A total of 247 key genes (GS > 0.7 and MM > 0.8) were selected from the turquoise module genes (Fig. 3F). The GO enrichment results revealed that these key genes were significantly enriched in the aging pathway (adjusted $p = 0.00006740$). Additionally, these genes were closely related to the extracellular matrix (ECM) and collagen (ECM organization, ECM structural constituent and its tensile strength, collagen-containing ECM, collagen trimer, and integrin binding), basement membrane, and immunity (myeloid leukocyte activation, cell chemotaxis, leukocyte migration, leukocyte chemotaxis, IgG binding, and immunoglobulin binding) (Fig. 3G). The KEGG pathway results with the minimum adjusted $p$-value also showed a significant relationship between key genes and ECM (ECM-receptor interaction) and immunity (complement and coagulation cascades and phagosome). Additionally, KEGG revealed that these key genes were associated with PI3K-Akt, AGE-RAGE in DM complications, and lipids (Fig. 3H). The minimum enrichment scores for GO and KEGG were 3.61 and 2.32, respectively.

## Immune-related features of DKD and co-differential DKD subtypes

GSEA of the GSE96804 gene set revealed that immune-related pathways were significantly enriched in DKD samples compared to those in control samples (Figs. 4A–4B). Immune cell infiltration analysis of the GSE96804 data using "CIBERSORTx" revealed significant differences in immune infiltration between the DKD and normal control samples (Fig. 4C). Memory B lymphocytes, CD8+T lymphocytes, macrophages (M0, 1 and 2), and resting mast cells were significantly increased in DKD compared to normal control samples.

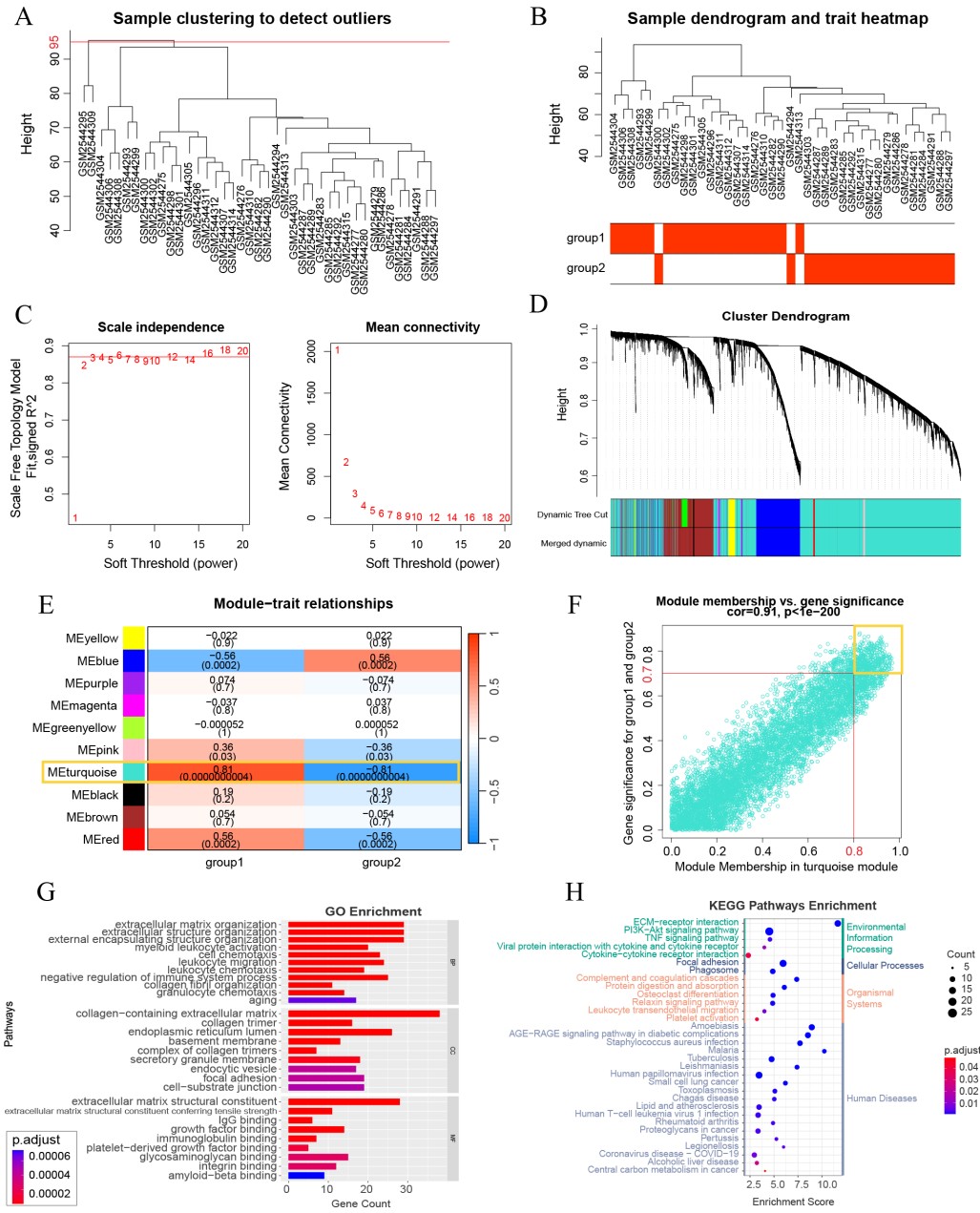

**Figure 3** **Weighted gene correlation network analysis (WGCNA) and key gene analysis.** (A) Sample clustering of 41 patients with DKD in the GSE96804 dataset. The two outliers shown in this figure were removed before the next step. (B) Clustering dendrogram of the remaining samples. Aging and DKD co-differential DKD subtypes are related to the sample dendrogram. (C) Analysis of network topology for various soft-thresholding powers. (D) Clustering dendrogram of genes of the remaining samples, together with the assigned original module colors and the final merged module colors. (E) Heatmap illustrations of the relationships between merged modules and traits of aging and DKD co-differential DKD subtypes. (F) Scatter plot of module membership (MM) and gene significance (GS) of the turquoise module. (G–H) Analysis of key WGCNA genes (in the turquoise module, and met the conditions of MM > 0.8 and GS > 0.7). (G) GO analysis results of key WGCNA genes. (H) KEGG enrichment analysis of key WGCNA genes.

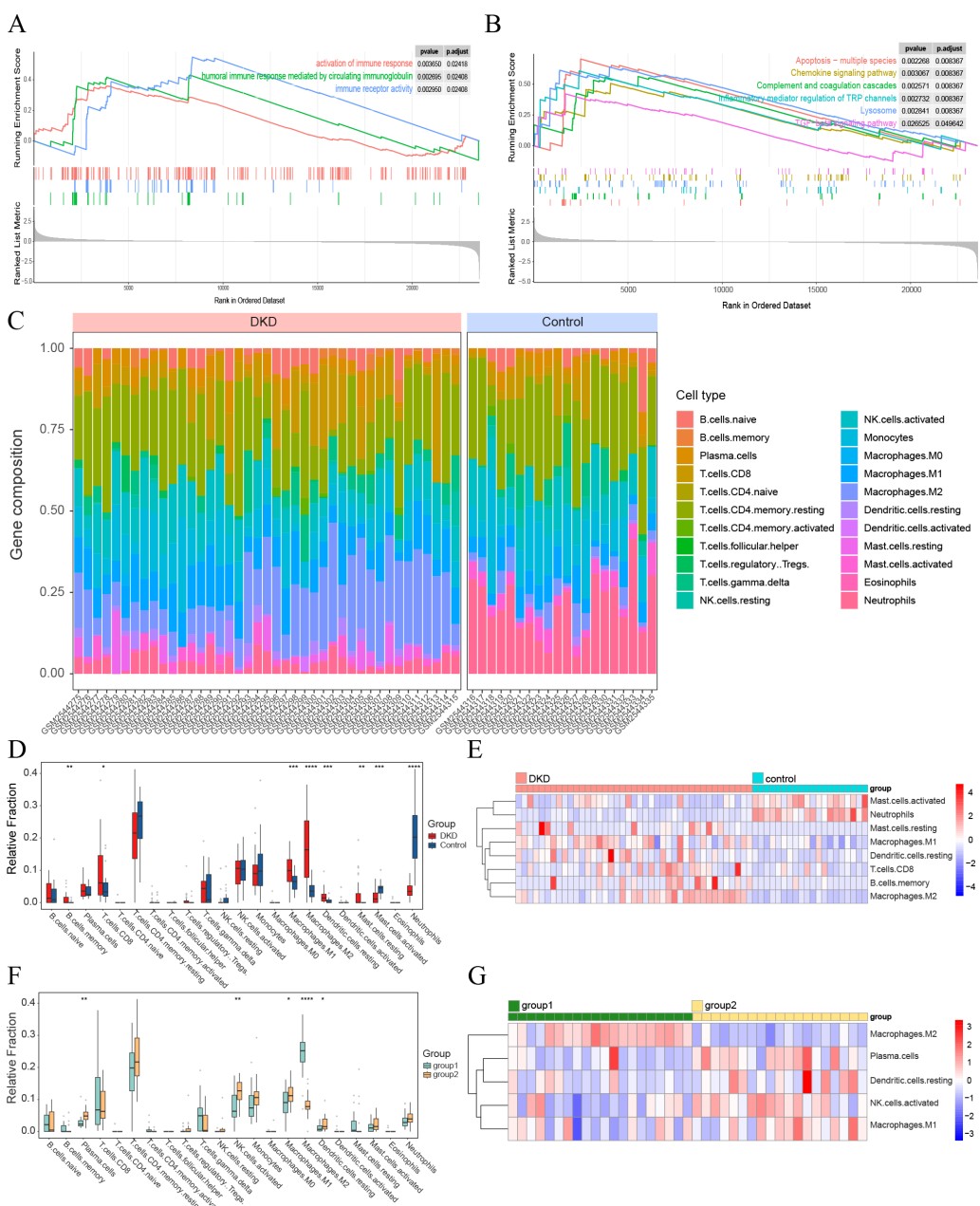

**Figure 4** **Visualization of immune-related analyses.** (A–B) GSEA results of all GSE96804 gene sets. (A) Immune-related results of KEGG GSEA. (B) Immune-related results of GO GSEA. (C) The composition of 22 types of infiltrating immune cells in the DKD and control samples. (D) Box plots depicting the variances in immune cell infiltration in DKD and control samples. (E) Heatmap showing immune-infiltrating cells with significant differences between the DKD and control samples. (F) Box plots depicting the differences in immune cell infiltration between aging and DKD co-differential DKD subtypes (group 1 *vs.* group 2). (G) Heatmap showing immune-infiltrating cells with significant differences between aging and DKD co-differential DKD subtypes (group 1 *vs.* group 2). Red and blue represent up- and downregulation, respectively. $^{*}p < 0.05$; $^{**}p < 0.01$; $^{***}p < 0.001$; $^{****}p < 0.0001$, Student's *t*-test.

Activated mast cells and neutrophils showed a significant decrease in DKD (Figs. 4D–4E). Immune cell infiltration differed significantly between the aging and DKD co-differential DKD subtypes. The number of plasma cells, M1, activated natural killer (NK) cells, and dendritic cells in the resting phase was reduced in group 1 compared with group 2, while macrophage M2 was markedly increased (Figs. 4F–4G).

## Analysis of significant co-differential genes associated with DKD and aging

The intersection of 644 significant DKD-DEGs (FDR < 0.05 and |LogFC| > 1) and 307 aging-related genes resulted in the identification of 15 genes (Fig. 5A). These genes (FOS, HSPA1A, HSPA1B, MXD1, JUN, PTGS2, EGR1, PCK1, EGF, CETP, KL, SST, IGF1, SLC13A1, and PLCG2) were identified as significant co-differential genes associated with DKD and aging (Table 2). The expression levels of all genes were significantly downregulated (LogFC < −1) (Figs. 5B–5C). PPI network analysis indicated a close interaction between the proteins encoded by DKD and aging significant co-differential genes (Fig. 5D). Furthermore, Pearson's correlation test revealed strong positive correlations between DKD and aging significant co-differential genes (Fig. 5E). In addition, KEGG (Fig. 5F) and GO (Fig. 5G) enrichment analysis revealed that DKD and aging significantly co-differential genes were involved in the biological process of regulating stress (temperature, chemical stress, and oxidative stress), aging, glucolipid metabolism, and glycosylation (AGE-RAGE pathway in DM complications, lipid and atherosclerosis, and response to fructose and lipopolysaccharide), as well as immune-related signaling pathways (apoptosis, TNF, MAPK, IL-17, Ras, NF-κB, Toll-like receptor, PI3K-Akt, and T cell and B cell receptor). The minimum enrichment scores for KEGG and GO enrichment analyses of DKD and aging significant co-differential genes were 9.9 and 5.31, respectively.

## Development of a genetic diagnostic model for DKD

The LASSO method was used to screen for significant co-differential genes associated with DKD and aging. Based on this, a genetic diagnostic model with five co-differential genes (FOS, HSPA1A, IGF1, CETP, and PCK1) was developed (Figs. 6A–6C). The AUC value of the validation set was 0.876 (Fig. 6D), indicating that the model had an excellent diagnostic performance. The box plots showed the predicted values of the LASSO model for DKD and normal controls separately in the training and validation sets (Figs. 6E–6F), demonstrating the strong discrimination ability of this model.

## Annotation and analysis of cluster subtypes of single-cell data

After screening the scRNA-seq data of GSE131882, there were 16,504 cells with an average of 1,301 genes and 13,726 counts per cell in three kidney samples from early DKD patients and three from normal controls. UMAP analysis revealed significant changes in single-cell data between patients with early DKD and controls (Fig. 7A). Eleven renal cell types and leukocytes were annotated based on renal cell type-specific DEGs (Figs. 7B–7C, Table 3). Subsequently, the four leukocyte subtypes were classified using marker genes (Table 3). The classification results suggested that only one T lymphocyte and six monocytes were present in control kidney samples. Meanwhile, 47 T lymphocytes, 24 B lymphocytes, 31

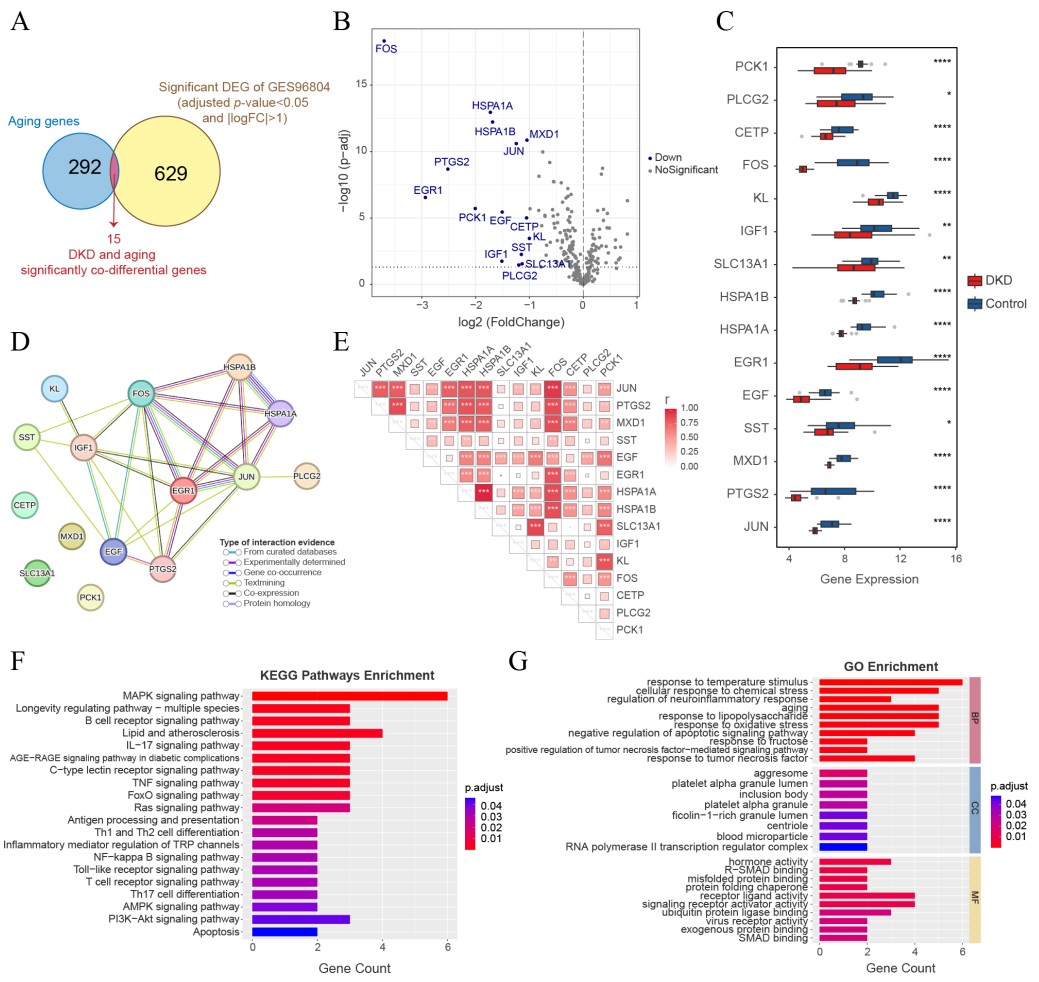

**Figure 5** **Analysis of DKD and aging significant co-differential genes.** (A) Venn diagram showing the significant DEGs (for DKD and control) intersecting with aging-related genes. (B) Volcano plot only shows aging-related gene expression in GSE96804; blue dots display DKD and aging significantly co-differential genes (above the dotted line, FDR <0.05). (C) Box plots of DKD and aging significant co-differential genes. $*p < 0.05$; $**p < 0.01$; $***p < 0.001$; $****p < 0.0001$, Student's $t$-test. (D) Protein-protein interaction (PPI) network analysis of proteins encoded by DKD and aging significant co-differential genes. (E) Pearson's correlation between DKD and aging significant co-differential genes in DKD samples. The larger the squares, the greater is the r-value (Pearson's correlation coefficient). $*p < 0.05$; $**p < 0.01$; $***p < 0.001$. (F–G) Kyoto Encyclopedia of Genes and Genomes (KEGG) pathway and Gene Ontology (GO) analysis results of biological processes (BP), cellular components (CC), and molecular functions (MF) enriched by DKD and aging significant co-differential genes.

monocytes, and 14 plasma cells were detected in the early DKD kidney samples (Fig. 7D). These findings demonstrated that immune cells were significantly increased in patients with early DKD.

## Changes in expression of model genes in single-cell RNA-Seq data

The expression of aging-related diagnostic model genes (FOS, HSPA1A, IGF1, CETP, and PCK1) was analyzed in all cells from the kidney samples. The results revealed that the

**Table 2  Diabetic kidney disease (DKD) and aging co-differential genes.**

| Abbreviation | Full name |
| --- | --- |
| CETP | Cholesteryl ester transfer protein |
| EGF | Epidermal growth factor |
| EGR1 | Early growth response 1 |
| FOS | Fos proto-oncogene |
| HSPA1A | Heat shock protein family A (Hsp70) member 1A |
| HSPA1B | Heat shock protein family A (Hsp70) member 1B |
| IGF1 | Insulin like growth factor 1 |
| JUN | Jun proto-oncogene |
| KL | Klotho |
| MXD1 | MAX dimerization protein 1 |
| PCK1 | Phosphoenolpyruvate carboxykinase 1 |
| PLCG2 | Phospholipase C gamma 2 |
| PTGS2 | Prostaglandin-endoperoxide synthase 2 |
| SLC13A1 | Solute carrier family 12 member [sodium/chloride] 3 |
| SST | Somatostatin |

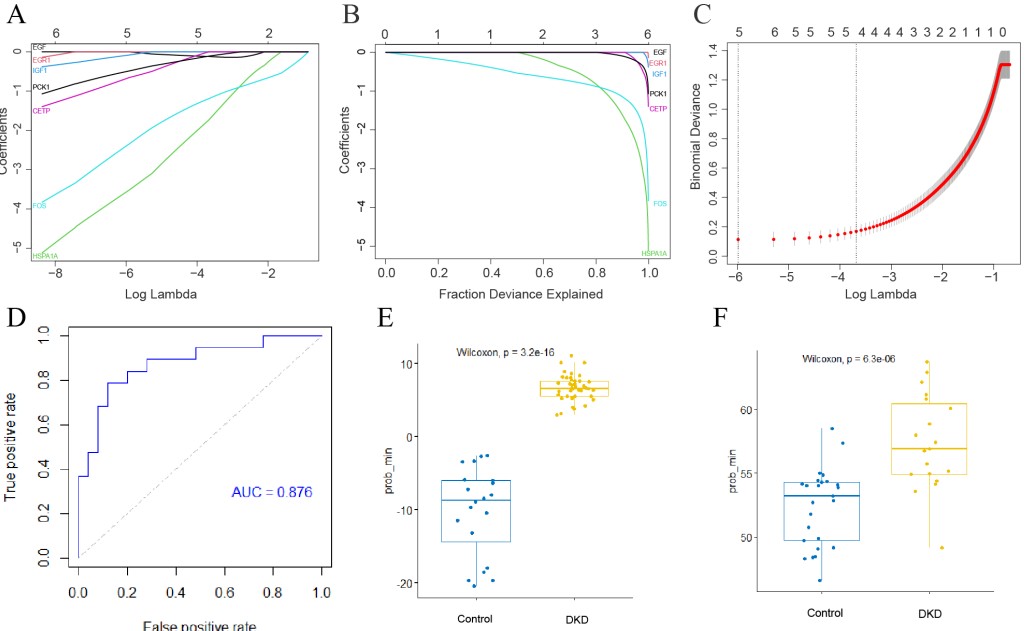

**Figure 6  Establishment of DKD genetic diagnostic model.** (A–C) LASSO regression analysis to screen for better DKD and aging co-differential genes for DKD diagnosis. (D) ROC curves for assessing the diagnostic efficacy of the LASSO model in GSE30528 and GSE30529 datasets. (E) Box diagram of the LASSO diagnostic model-predicted values in the GSE96804 dataset. (F) Box diagram of the LASSO diagnostic model-predicted values in the GSE30528 and GSE30529 datasets.

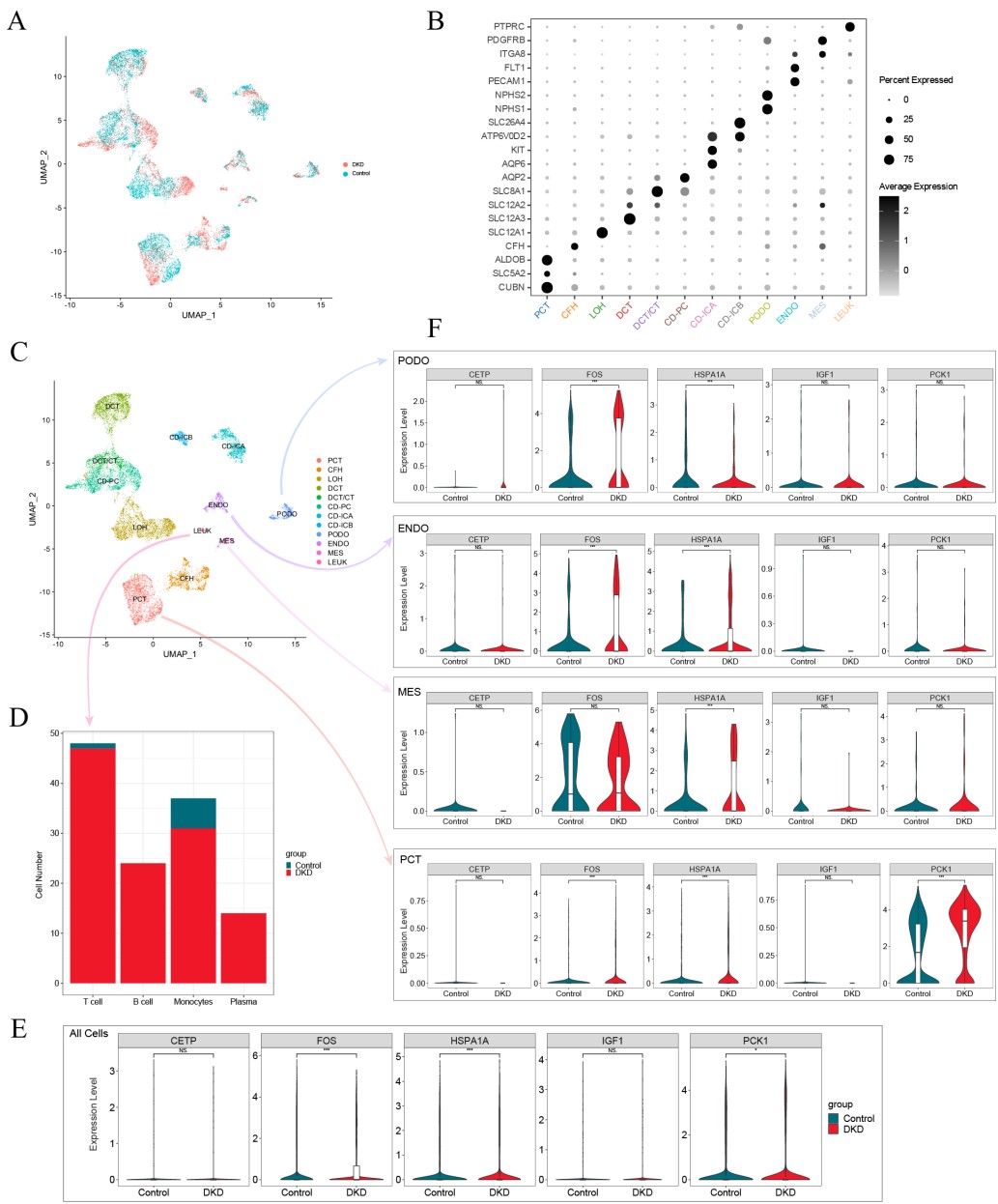

**Figure 7 Single-cell analysis of patients with early DKD.** (A) UMAP plots of comparisons between early DKD patients and control samples. (B) Cell clusters are recognized by renal cell type-specific DEGs. (C) UMAP plots of cell clusters. (D) Stacked bar graph of immune cell subtypes in early DKD and control groups. (E) Violin plot of the expression of five DKD diagnostic model genes in all cells. (F) Violin plot of five DKD diagnostic model gene expression in podocytes (PODO), endothelium (ENDO), mesangial cell (MES), and proximal convoluted tubule (PCT) cell types.

differences in the gene expression of FOS, HSPA1A, and PCK1 between the early DKD and control groups were statistically significant (Fig. 7E). The above statistical analyses were also performed independently for podocyte (PODO), endothelium (ENDO), mesangial cell (MES), and proximal convoluted tubule (PCT) cell types (Fig. 7F). The results revealed

**Table 3** Cell-type-specific differentially expressed genes for cell types annotation of single cell data.

| Cell types | Abbreviations | Marker gene |
|---|---|---|
| Proximal convoluted tubule | PCT | CUBN, SLC5A, ALDOB |
| Complement factor H | CFH | CFH |
| Loop of Henle | LOH | SLC12A1 |
| Distal convoluted tubule | DCT | SLC12A3, SLC12A2 |
| Distal convoluted tubule/connecting tubule | DCT/CT | SLC8A1 |
| Collecting duct-principal cell | CD-PC | AQP2 |
| Collecting duct- intercalated cell A | CD-ICA | AQP6, KIT, ATP6V0D2 |
| Collecting duct- intercalated cell B | CD-ICB | SLC26A4 |
| Podocyte | PODO | NPHS1, NPHS2 |
| Endothelium | ENDO | PECAM1, FLT1 |
| Mesangial cell | MES | ITGA8, PDGFRB |
| Leukocyte | LEUK | PTPRC |
| T lymphocyte | T_cell | CD247, CD96, CD28 |
| Monocytes | Monocytes | FCGR2A, CSF1R, CSF2RA, CD86 |
| B lymphocyte | B_cell | MS4A1, PAX5 |
| Plasma cells | Plasma | CD38, SDC1 |

that FOS expression increased significantly in PODO of early DKD. In addition, HSPA1A levels significantly decreased. The expression of FOS and HSPA1A in ENDO was markedly increased in early DKD. In MES, the expression of HSPA1A increased in early DKD. FOS, HSPA1A, and PCK1 expression in PCT was significantly higher in early DKD patients than in controls. These findings suggested that FOS, HSPA1A, and PCK1 may affect early DKD injury.

## Histopathology of the DKD and control mice

The DKD mouse model (Fig. 8A) was established by injecting STZ and feeding the mice a high-fat diet. In the kidney histology section of DKD mice, some pathological alterations were observed in the kidney histological section of DKD mice (Fig. 8B). Hematoxylin and eosin staining revealed diffuse mesangial expansion, including mesangial cell proliferation, mesangial matrix hyperplasia, nodular changes, and tubular cell hypertrophy. Masson's Trichrome staining revealed a significant quantity of blue-stained collagen accumulated in the glomeruli, which suggested fibrosis. PAS staining revealed glycogen deposition, mesangial cell proliferation, and thickened basement membrane in the glomeruli.

## Expression of diagnostic model genes in DKD mouse model

Four of the five genes in the aging-related diagnostic model (Igf1, Hspa1a, Fos, and Pck1) were identified in mice. The mean expression of all these genes was reduced in DKD mice compared with that in the control, and the reduction in Igf1 expression was statistically significant (Fig. 8C, Table S1).

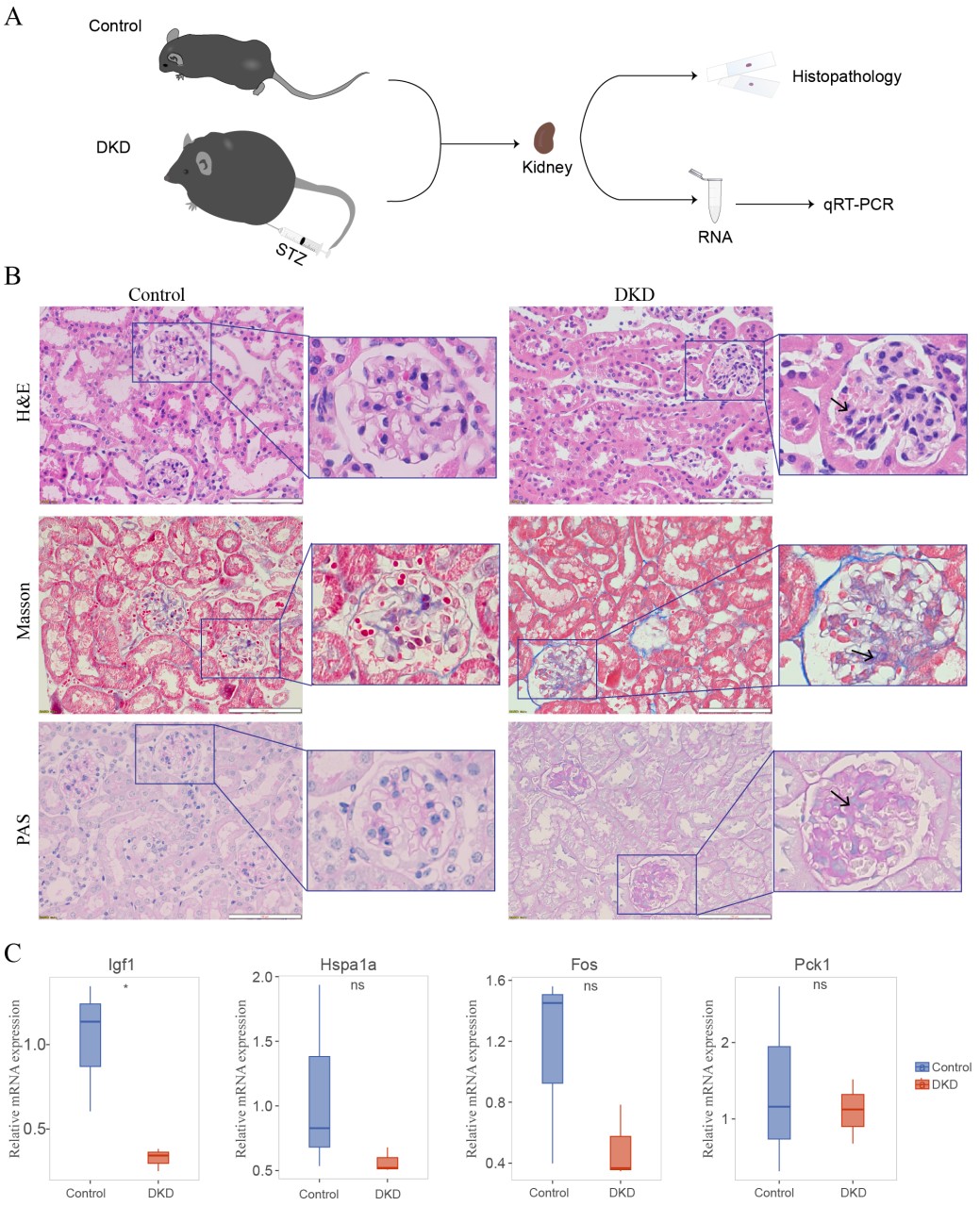

**Figure 8** **Mouse experiments and results.** (A) Schematic of the experimental process of the DKD mouse model and control mice. (B) Kidney histological sections from DKD and normal control mice stained with Hematoxylin and Eosin (H&E), Masson's Trichrome, and Periodic Acid-Schiff (PAS). Diffuse mesangial expansion with mesangial cell proliferation, mesangial matrix hyperplasia, glomerular hypertrophy, nodular changes, and collagen deposition can be seen in the kidneys of DKD mice. (C) qRT-PCR results of DKD mice compared with those of control mice. No significance (ns); $*p < 0.05$, Student's $t$-test (applied in most of the data) or Mann–Whitney test (only applied to Hspa1a data).

## DISCUSSION

In this study, DKD and aging co-differential genes and DKD and aging significant co-differential genes were comprehensively analyzed. Based on these genes, a diagnostic model for DKD was developed.

Aging is a significant risk factor for DKD (*Alicic, Rooney & Tuttle, 2017*). High glucose stimulation has been shown to accelerate cellular senescence in several types of kidney cells (*Cao et al., 2019*; *D'Onofrio et al., 2016*; *Kitada et al., 2014*; *Tsai et al., 2018*; *Verzola et al., 2008*). Through whole-gene GSEA, GO, and KEGG enrichment analyses, we found that aging-related pathways were significantly enriched, demonstrating that aging plays an important role in the pathogenesis and progression of DKD at the genetic level. We established aging and DKD co-differential DKD subtypes based on heterogeneity in the expression of DKD and aging co-differential genes. Subsequently, we observed that different aging and DKD co-differential DKD subtypes had significant differences in whole-genome expression and biological processes closely associated with both aging and DKD development, such as glucolipid metabolism and epithelial-mesenchymal transition. Furthermore, subtype categorization based on co-differential genes yielded results similar to those of the whole-gene-based WGCNA dendrograms. These findings suggest that integrative research on aging and DKD is important for the development of novel diagnostic and therapeutic strategies.

Through an integrative study of aging and DKD, co-differential genes were obtained by gene intersection. These DKD and aging co-differential genes, as well as DKD and aging significant co-differential genes, are involved in not only aging, but also various biological processes, including stress, apoptosis, glucolipid metabolism, glycosylation, and several immune-related signaling pathways. These biological processes are similar to those involved in the pathophysiology of DKD. This suggests that these genes may play a decisive role in DKD progression and could serve as novel markers for DKD diagnosis.

Therefore, a genetic diagnostic model for DKD has been developed based on these genes. ROC curves demonstrated the excellent diagnostic efficiency of this model. The expression of these model genes was validated using single-cell sequencing data and mouse models. In our DKD mouse model, kidney histopathology revealed increased glycogen deposition, mesangial proliferation, thickened basement membrane, and increased fibrous collagen, all of which are pathological hallmarks of DKD (*Alicic, Rooney & Tuttle, 2017*). The expression levels of FOS, HSPA1A, PCK1, and IGF1 differed between the two assays.

In line with our findings, previously published studies have shown that FOS, HSPA1A, PCK1, and IGF1 are directly or indirectly associated with DKD. FOS and HSPA1A are closely associated with immunity.

IGF1, which encodes insulin-like proteins and activates the type I IGF receptor, is a component of the growth hormone-IGF-somatostatin system that plays a critical role in DKD (*Segev et al., 2004*). Research has indicated that IGF1 can inhibit apoptosis and DNA damage in high glucose-stimulated mesangial cells (*Kamenický et al., 2014*). Consistent with our study, significantly decreased serum IGF1 and decreased renal Igf1 RNA levels were found in db/db mice with DKD in a previous study (*Segev et al., 2007*). A lifestyle

intervention trial also found that low IGF1 levels predicted DM onset in patients with prediabetes (*Meyer et al., 2022*). However, unlike our study, which demonstrated low IGF1 expression in DKD patients, some studies have reported that IGF expression increases with DKD progression (*Dong et al., 2019*). Elevated IGF-1 levels may be associated with adverse renal outcomes in patients with DKD (*Mohebi et al., 2023*). However, studies have also demonstrated that IGF1 expression varies with stage of DM (*Mohebi et al., 2023*). Moreover, both low and high normal levels of IGF-I may be associated with insulin resistance (*Friedrich et al., 2012*). These results may explain the inconsistent data obtained from different experiments. At the same time, these findings also suggest that the treatment strategy for DKD patients with high IGF1 gene expression should differ from those with low IGF1 gene expression.

PCK1, a critical regulator of gluconeogenesis, is a key regulator of renal tubular cell metabolism that can alter mitochondrial function (*Verissimo et al., 2023*). Furthermore, PCK1 shows significantly higher expression in the kidneys than in other organs (*Fagerberg et al., 2014*). Consistent with our DEG analysis and mouse qPCR results, previous studies have shown that Pck1 expression is downregulated in STZ-induced DKD mice. Moreover, overexpression of Pck1 has been found to improve proteinuria and collagen deposition (*Hasegawa et al., 2023*). These results suggest that PCK1 plays a protective role in DKD and could be a potential diagnostic and therapeutic target for future studies. In contrast, some researchers have reported that Pck1 mRNA is significantly increased in the kidneys of DKD db/db mice (*Watanabe et al., 2018*). This finding corroborated the conclusion of our single-cell analysis of patients with early DKD, revealing that Pck1 expression levels may vary at different stages of DKD, which requires further investigation.

FOS encodes c-Fos, a transcription factor subunit of activator protein 1 (AP-1). AP-1 is one of the targets of the MAPK signaling cascade (*Atsaves et al., 2019*) and can be activated by various DKD-related stimuli (*Huang et al., 2014*). Furthermore, AP-1 can affect inflammation through different mechanisms, such as regulating naive T cell differentiation and regulating the activity of the innate immune system (*Wagner & Eferl, 2005*). Furthermore, some studies investigations have shown that AP-1 can also affect renal fibrosis (*Huang et al., 2014*).

HSPA1A encodes heat shock protein 70 (HSP70), a type of molecular chaperone involved in protein folding and remodeling (*Rosenzweig et al., 2019*). It is also effective as an immunological adjuvant. Hsp70 can directly inhibit heat shock factor 1, which is a vital component of the heat shock response. Moreover, intracellular Hsp70 downregulates NF-κB activation and exhibits immunosuppressive activity. When Hsp70 is extracellular, the reverse effect occurs (*Tukaj, 2020*). Previous research has indicated that serum HSP70 levels are higher in patients with DM with albuminuria than in those without albuminuria (*Nargesi et al., 2016*), which is contrary to the trend of decreased HSPA1A gene expression in the kidneys of DKD patients that we investigated. Intracellular and extracellular HSP70 play distinct roles in inflammation (*Zhou et al., 2023*), which may explain the different trends in the kidney compared to the serum. However, in our single-cell analysis of patients with early DKD, the expression of HSPA1A in endothelial, mesangial, and proximal renal tubular epithelial cells increased, whereas podocyte HSPA1A expression decreased. Other studies

have shown contradictory results, suggesting that both intracellular and extracellular Hsp70 play a dual role in inflammation (*Tukaj, 2020*). Its complex and confusing relationship with DKD warrants further investigation.

In contrast, studies have revealed that DKD shows a low-level of chronic inflammation (*Pérez-Morales et al., 2019*). Immune disorders in patients with DKD were observed in our study. Whole-gene GSEA also revealed significant gene enrichment in immune-related pathways in patients with DKD compared to controls. Furthermore, immune cell infiltration analysis revealed significant heterogeneity between DKD and normal tissues. Similarly, scRNA-seq data analysis revealed that early DKD patients had significantly higher levels of T lymphocytes, B lymphocytes, monocytes, and plasma cells than controls. Consistent with our findings, many studies have demonstrated a promoting effect of immunity on DKD development. Diabetic stress recruits leukocytes, such as macrophages and mast cells, *via* danger-associated molecular patterns, resulting in kidney damage (*Tesch, 2017*). T cells, which are recruited to the kidneys of patients with DKD, also contribute to the development of DKD (*Chen et al., 2022*).

Aging also causes low-level chronic inflammation (*Barbé-Tuana et al., 2020*), and senescent cells may trigger an immunological response to DKD. Our findings revealed that different aging and DKD co-differential DKD subtypes exhibited significant immunological heterogeneity. ssGSEA revealed differences in genes encoding inflammatory responses between these subtypes. Furthermore, a noticeable discrepancy in the immune cell infiltration was observed. In addition, enrichment analysis of key WGCNA genes revealed that these genes had a significant biological role in DKD pathogenesis and aging, as well as a close relationship with immunity and inflammation.

These findings suggest that DKD progression may be linked to immunosenescence and that aging plays a significant role in DKD immunological disorders. Moreover, aging-targeted therapy may represent a novel treatment for immunological dysfunction in patients with DKD. Our diagnostic model genes (IGF1, CETP, PCK1, FOS, and HSPA1A), particularly IGF1, PCK1, FOS, and HSPA1A, are potential targets.

Our study has several limitations. First, our findings were based on analysis of the mRNA expression levels of genes rather than protein levels. Second, owing to publication bias of available data, a small sample size, and limitations of animal models, we plan to conduct experiments in subsequent studies to validate the conclusions and address these limitations.

## CONCLUSIONS

This study identified the co-differential genes between DKD and aging, and developed a DKD diagnostic model based on these genes. Our diagnostic model genes (IGF1, CETP, PCK1, FOS, and HSPA1A), particularly IGF1, PCK1, FOS, and HSPA1A, are potential diagnostic and therapeutic targets for DKD treatment. Additionally, DKD progression may be linked to immunosenescence.

## ACKNOWLEDGEMENTS

We thank Shanghai NewCore Biotechnology Co., Ltd., for providing visualization support. Besides, we thank Bullet Edits Limited for the linguistic editing and proofreading of the manuscript.

### Funding

This work was supported by the Lanzhou Science and Technology Bureau Talent Innovation Project (2021-RC-94 and 2014-RC-64), the National Natural Science Foundation of China (No. 81960142), the Youth Science and Technology Fund Program of Gansu Province (No. 21JR1RA157), and the Lanzhou University Second Hospital Youth Fund (CY2021-QN-B01) and Project of Department of Education of Gansu Province (2022B-050). Our experiments are supported by the Clinical Medical Research Center of Gansu Province (21JR7RA436). The funders had no role in study design, data collection and analysis, decision to publish, or preparation of the manuscript.

### Grant Disclosures

The following grant information was disclosed by the authors:
Lanzhou Science and Technology Bureau Talent Innovation Project: 2021-RC-94, 2014-RC-64.
National Natural Science Foundation of China: 81960142.
Youth Science and Technology Fund Program of Gansu Province: 21JR1RA157.
Lanzhou University Second Hospital Youth Fund: CY2021-QN-B01.
Project of Department of Education of Gansu Province: 2022B-050.
Clinical Medical Research Center of Gansu Province: 21JR7RA436.

### Competing Interests

The authors declare there are no competing interests.

### Author Contributions

- Hongxuan Du conceived and designed the experiments, performed the experiments, analyzed the data, prepared figures and/or tables, authored or reviewed drafts of the article, and approved the final draft.
- Kaiying He performed the experiments, analyzed the data, prepared figures and/or tables, animal breeding, and approved the final draft.
- Jing Zhao performed the experiments, prepared figures and/or tables, data collection, and approved the final draft.
- Qicai You performed the experiments, prepared figures and/or tables, animal breeding, and approved the final draft.
- Xiaochun Zhou performed the experiments, prepared figures and/or tables, animal breeding, and approved the final draft.
- Jianqin Wang conceived and designed the experiments, authored or reviewed drafts of the article, and approved the final draft.

## Animal Ethics

The following information was supplied relating to ethical approvals (i.e., approving body and any reference numbers):

The animal experiment passed the ethical review of experimental animals in the Second Hospital of Lanzhou University, No D2019-149.

## Data Availability

The RT-qPCR Raw measurements can be found in ''supplementary files 1.pdf'' and "Raw data of RT-qPCR.xlsx''. The raw data shows the expression of genes in DKD mice kidney compared with Control. The code and related data can be found in "Code.R" and "data referenced by the code.zip''.

## Supplemental Information

Supplemental information for this article can be found online at http://dx.doi.org/10.7717/peerj.17046#supplemental-information.

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
