# Peer review of "Co-differential genes between DKD and aging: implications for a diagnostic model of DKD"

_PeerJ, doi:10.7717/peerj.17046_

## Round 0.1 · original submission · Major Revisions

Please carefully resolved all the reviewers' suggestions.

**Language Note:** The review process has identified that the English language must be improved. PeerJ can provide language editing services - please contact us at copyediting@peerj.com for pricing (be sure to provide your manuscript number and title). Alternatively, you should make your own arrangements to improve the language quality and provide details in your response letter. – PeerJ Staff

Reviewer 1 ·

Basic reporting

Du et. al., report an integrative study where they first identified the co-differential genes between diabetic kidney disease (DKD) and ageing using GEO dataset. Subsequently, developed a genetic model using regression analysis and feature extraction, and validated the expression these identified co-differential genes in early DKD patients using single cell RNA sequencing (scRNA-seq) and RT-qPCR test of DKD mice kidney. Based upon the findings, authors suggest that studying co-differential genes between DKD and ageing, especially genes FOS, HSPA1A, PCK1 and IGF1 provide potentially suitable path to diagnosis and treatment.
In general, the manuscript is adequately written and conveys crucial information regarding the relationship between aging and Diabetic Kidney Disease (DKD). However, there are several major concerns that need attention to enhance the manuscript:
1. The Abstract and Introduction lack clarity regarding the proposed connection between DKD and aging, as well as the overall aim of the study. It is unclear whether the authors aim to screen and propose a model studying DKD and aging as similar or parallel processes, or if it is an aging-related diagnostic model for DKD. This becomes clearer in the Results and Discussion section but should be addressed in the Abstract and Introduction.
2. The current title does not provide a clear and concise overview of the study's findings. A more suitable title, such as "Co-differential Genes Between DKD and Aging: Implications for a Diagnostic Model of DKD," should be considered to accurately reflect the study's overall aim.
3. The last two paragraphs of the Discussion section (lines 421-431) and the conclusion need sentence restructuring for clarity. Similar to point 1, these sections currently fail to clearly depict the exact aim and findings of the study.

Experimental design

a. The "peerj-91669-Code.R" file requires careful revision to correct syntax errors, especially at lines 107/108, 190/191, 232/233, 238/239, 390/391, and others.

b. Histological findings do not strongly indicate DKD development in mice. Masson and PAS stained images of DKD mice do not reveal visible mesangial proliferation or thickening of the basement membrane compared to control images. Adequate images with proper labeling should be provided, and it should be addressed whether imaging was conducted adequately or if the histological model failed to reach the full DKD stage. Perhaps, a more suitable (susceptible) mice strain should have been used for DKD model such as the db/db mouse, most widely used rodent model of DKD, or C57BLKS/J mice which are more prone to develop features of DKD (Sembach et. al., 2020).

Validity of the findings

a. In the "peerj-91669-qPCR_VALIDATION" section, the presented melt curves should be correctly labeled as melt peaks. Additionally, the standard curve is inaccurately plotted and reflects opposite of “High Ct should correspond to low quantity, and low Ct to high quantity”. Rectifying these errors is essential.

Additional comments

a. Including brief demographic information (table) of the subjects, such as the age of the subjects in the GEO datasets used would be beneficial for readers to ascertain the factors like aging-related genes and aging itself.
b. While the authors have cited adequate and relevant references, there is an opportunity in the Introduction and Discussion to include additional references to strengthen the connections between DKD and aging.

·

Basic reporting

In this manuscript, Du et. al explore the relationship between Diabetic Kidney Disease (DKD) and aging/inflammation. While the background and context the authors provide highlights the relevance of their study, the methods they use to seek to test their hypothesis (DKD involves canonical aging and inflammatory pathways) are not straightforward and the results seem pre-determined from the outset. The authors used a reference set of aging-associated genes, but it is unclear to me how the genes they report are "co-differential" with DKD because they only used DKD disease data in their analysis.

The writing is acceptable, but there are multiple sentences which are confusing or poorly worded. The authors have described which methods and have admirably provided all necessary code to recapitulate the analysis, however the methods section is too vague and lacks the necessary details for the reader to quickly assess and understand their approach.

The authors describe the results from each of the different types of analysis they performed, but there is no theme or narrative connecting these different sections. As a result, the manuscript is more piecemeal than a complete study, leaving this reader to feel as though it less than the sum of its parts. A future revision will need to substantially address this shortcoming to ensure that the reader is able to follow the results and what conclusions can be drawn from them; the authors need to provide a greater context and rationale for each step in their analysis, rather than just listing the different sections.

Experimental design

Du et al. use existing bulk RNA-seq and single cell RNA-seq data from DKD patients and control subjects, and use common analysis pipelines to analyze the data. They provide the necessary code to recapitulate the analysis. However, at the outset, it appears the authors filter their analysis only to aging-associated genes. As a reader, I am left wondering whether DKD is enriched for aging-associated genes, but the authors do not show those results - only that some DKD differential genes are also associated with aging. An overlap of ~%40 aging genes being differential in DKD seems modest. Rather than only testing whether aging genes are differential, the authors should perform differential analysis on all genes (preferably using limma's TREAT test - testing relative to a threshold - since they want to filter on FDR p-value as well as logFC, which would be better than just the limma t-test and then post-filtering on logFC) and then determine whether aging and inflammatory geneset are enriched (for example, using the CERNO test in tmod package or the fast GSEA implementation in fgsea) to illustrate that it's not coincidence that aging and inflammatory genes are found in differential DKD genes.

For the development of LASSO-based genetic determination model for DKD, it seems as though the authors pre-determined that only aging-related genes were used as an input for LASSO. This makes me wonder whether there would be other non-aging genes that would perform better in this scenario? This does not show that aging-related genes are the top predictors for genetic determination of DKD, just that there are 5 aging-associated can be used. How do they perform relative to other non-aging related genes? There's clearly a WGCNA cluster (turquoise) that contains aging-associated genes, so they may perform well in comparison with all genes, but it's not clear from the author's analysis. Enrichment of the turquoise WGCNA cluster with the aging-associated geneset could be used to show that there is a strong enrichment of aging genes in this particular cluster.

Validity of the findings

The results that the authors have presented are valid, the question I have is whether they support the claims that they make. Figures could be rearranged to improve the flow of the manuscript's narrative; for example, figure 4 could be placed after figure 1 (assuming that figure 1 is expanded to included all genes - not just aging-related ones). Figure 4 is the strongest evidence that there are indeed aging-related genes associated with DKD, but many of the other figures start from the assumption that aging-related genes are important a priori rather than showing an unbiased result that aging-related genes are enriched and then continuing down the investigate path towards this.

I appreciate that the authors have included a mouse model for DKD for validation; additional histopathological images could be provided for replication to ensure reader's confidence in the repeatability, as well as a clearer description of replicates used for the DKD mouse experiments.

Additional comments

The authors pose an interesting hypothesis and clearly performed thoughtful analysis on the data, however the manuscript is lacking in its narrative, formatting, detail, and description. The content of a suitable paper is there, however additional analysis of the data and substantial reworking of the manuscript is needed in order for it to be suitable for publication.

Reviewer 3 ·

Basic reporting

The article is fairly well-written, but it could benefit from editing to clarify the English for the reader. For example, just in the abstract, line 26 should read "and closely related" to aging. Line 29 should read, "the features of these genes were analyzed." Line 30, "Expression" should not be capitalized, etc.

Experimental design

Since the authors are conducting a meta-analysis, they are not making experimental design choices and have to instead rely on the choices made by the original studies. With that said, they combine data from several different sources, including microarray data, single cell RNA-seq data and RT-PCR to test their research question.

Validity of the findings

As stated above, the authors combine data from several different sources to support their claims. The datasets are fairly large as well, which improves the significance. For example, GSE96804 had 41 DM patients and 20 controls.

Additional comments

The authors present a meta-analysis of publicly-available data to look at the relationship between diabetic kidney disease (DKD) and aging. They use microarray, single cell RNA-seq, RT-PCR and mouse histology data to identify genes involved in DKD and aging and develop a diagnostic model. They show that the immune cell profile is different between DKD patients and controls and that several of the genes they identified were also differentially expressed in mice. Overall, the manuscript is a tour-de-force of bioinformatics supplemented with good molecular biology and cellular biology. The authors should be applauded for their efforts, however, there are some concerns that should be addressed to help the reader understand the author's work.

1) In general, the figures have low resolution and are often difficult to read. The authors and journal should work together to ensure every figure is legible.

2) In general, the manuscript is well-written and easy to follow, but as mentioned above, it could be significantly improved with editing of the English.

3) In the methods, the authors describe using 307 genes from the aging database and testing for significant differences using Limma in the DKD and aging datasets. All statistical methods are built on assumptions about the data and it may be problematic to just use Limma on 307 genes as it may have been designed to test significance using data from the entire array. The volcano plot in Figure 1A suggests Limma was run on the whole dataset though, so this may not have happened. The testing protocol should be clarified, in that case.

4) In figure 1c, the legend does not describe the meaning of the number of lines, the color of the lines or the color of the circles.

5) In figure 1d, the legend does not describe the meaning of the boxes, as some appear to have 2 stars and some have 3 stars. Also, what does the size of the box signify?

6) The legend is cutoff in Figure 1e.

7) What dataset are the authors using for the pathway analysis? On line 223, the authors state that the 159 genes were the "DKD and aging co-differential genes". Line 226 states that the 15 genes were the "DKD and aging significantly co-differential genes." Line 232 says they used the "DKD and aging significantly co-differential genes" to run the pathway analysis, which suggests they used the 15 gene list. However, it is very rare to find significantly enriched pathways with such a small gene list. Did they actually use the 159 gene list? That would make more sense.

8) What enrichment score are the authors considering as significant for their pathways analyses? Generally investigators report 2-fold enrichment and above, but it can vary and should be described.

9) In figure 2G, one would assume red means up-regulated and blue is down-regulated, but this should be clarified.

10) In figure 3 C and E, up- and down-regulated colors need to be clarified.

11) In figure 4, the authors describe selecting 247 "key genes". They need to explain how this was done and what it means to be a "key gene".

12) For Figure 4G and H, the reader again needs to be informed about the enrichment score cutoff used.

13) Figure 5- the color of the lines is not defined.

14) In the methods, the authors state that the "average values of probes were computed" when more than one probe was available for a given gene. While this is a reasonable approach and I don't think the authors need to change anything for this manuscript, in the future they may want to look more carefully at each probeset as many are known not to measure the gene in question at all. Eliminating these poor performing probesets will likely significantly improve their results in future studies.

---

## Round 0.2 · accepted · Accept

There are no further comments.

Reviewer 1 ·

Basic reporting

I appreciate authors for their meticulous revision based upon the editor's and reviewers' feedback. The revised version of the manuscript has undergone substantial improvements, effectively communicating its aims and findings in a compelling manner. I don't have any critical concerns at this point.

Experimental design

no comment

Validity of the findings

no comment

Additional comments

no comment

Reviewer 3 ·

Basic reporting

The English has improved.

Experimental design

The authors have addressed my concerns.

Validity of the findings

The authors have addressed my concerns.

Additional comments

None